# The dynamic three-dimensional organization of the diploid yeast genome

**Seungsoo Kim[1], Ivan Liachko[1], Donna G Brickner[2], Kate Cook[1], William S Noble[1], Jason H Brickner[2], Jay Shendure[1,3]\*, Maitreya J Dunham[1]\***

[1]Department of Genome Sciences, University of Washington, Seattle, United States; [2]Department of Molecular Biosciences, Northwestern University, Evanston, United States; [3]Howard Hughes Medical Institute, University of Washington, Seattle, United States

**Abstract** The budding yeast *Saccharomyces cerevisiae* is a long-standing model for the three-dimensional organization of eukaryotic genomes. However, even in this well-studied model, it is unclear how homolog pairing in diploids or environmental conditions influence overall genome organization. Here, we performed high-throughput chromosome conformation capture on diverged *Saccharomyces* hybrid diploids to obtain the first global view of chromosome conformation in diploid yeasts. After controlling for the Rabl-like orientation using a polymer model, we observe significant homolog proximity that increases in saturated culture conditions. Surprisingly, we observe a localized increase in homologous interactions between the *HAS1-TDA1* alleles specifically under galactose induction and saturated growth. This pairing is accompanied by relocalization to the nuclear periphery and requires Nup2, suggesting a role for nuclear pore complexes. Together, these results reveal that the diploid yeast genome has a dynamic and complex 3D organization.

**\*For correspondence:** shendure@uw.edu (JS); maitreya@uw.edu (MJD)

**Competing interests:** The authors declare that no competing interests exist.

## Introduction

The genome is actively organized in the nucleus in both space and time, and this organization impacts fundamental biological processes like transcription, DNA repair, and recombination (*Taddei et al., 2010*). The budding yeast *S. cerevisiae* has been a useful model for studying eukaryotic genome conformation and its functional implications (*Taddei et al., 2010*). The predominant feature of yeast 3D genome organization is its Rabl-like orientation (*Jin et al., 1998*) (*Figure 1A*): during interphase, the centromeres cluster at one end of the nucleus, attached to the spindle pole body, and chromosome arms extend outward toward the nuclear periphery where the telomeres associate (*Schober et al., 2008*; *Therizols et al., 2010*), like in anaphase. In addition, the ribosomal DNA array forms the nucleolus, opposite the spindle pole (*Yang et al., 1989*), splitting chromosome XII into two separate domains that behave as if they were separate chromosomes. This organization largely persists through the cell cycle (*Jin et al., 1998*) and even in stationary phase, albeit with increased telomere clustering and decreased centromere clustering (*Guidi et al., 2015*; *Rutledge et al., 2015*).

Genome-wide chromosome conformation capture methods like Hi-C have both confirmed these microscopy observations and permitted systematic analyses of the functional clustering of genomic elements like tRNA genes and origins of replication (*Duan et al., 2010*). However, multiple studies have argued that a simple volume-exclusion polymer model of chromosomes in a Rabl-like orientation is sufficient to explain microscopy and Hi-C data of the budding yeast genome (*Tjong et al., 2012*; *Wong et al., 2012*), at least in haploids grown under standard lab conditions. These studies have argued that even the functional clustering that is observed may simply be a consequence of

**eLife digest** Most of the DNA in human, yeast and other eukaryotic cells is packaged into long thread-like structures called chromosomes within a compartment of the cell called the nucleus. The chromosomes are folded to fit inside the nucleus and this organization influences how the DNA is read, copied, and repaired. The folding of chromosomes must be robust in order to protect the organism's genetic material and yet be flexible enough to allow different parts of the DNA to be accessed in response to different signals.

A biochemical technique called Hi-C can be used to detect the points of contact between different regions of a chromosome and between different chromosomes, thereby providing information on how the chromosomes are folded and arranged inside the nucleus. However, most animal cells contain two copies of each chromosome, and the Hi-C method is not able to distinguish between identical copies of chromosomes. As such, it remains unclear how much the chromosomes that can form pairs actually stick together in a cell's nucleus.

Unlike humans and most organisms, two distantly related budding yeast species can mate to produce a "hybrid" in which the chromosome copies can easily be distinguished from each other. Kim et al. now use Hi-C to analyze how chromosomes are organized in hybrid budding yeast cells.

The experiments reveal that the copies of a chromosome contact each other more frequently than would be expected by chance. This is especially true for certain chromosomal regions and in hybrid yeast cells that are running out of their preferred nutrient, glucose. In these cells, the regions of both copies of chromosome 13 near a gene called *TDA1* are pulled to the edge of the nucleus, which helps the copies to pair up and the gene to become active. The protein encoded by *TDA1* then helps turn on other genes that allow the yeast to use nutrients other than glucose.

Many questions remain about how and why DNA is organized the way it is, both in yeast and in other organisms. These findings will help guide future experiments testing how the two copies of each chromosome pair, as well as what purpose, if any, this pairing might serve for the cell. A better understanding of the fundamental process of DNA organization and its implications may ultimately lead to improved treatments for genetic diseases including developmental disorders and cancers.

the Rabl-like orientation coupled with biases in the chromosomal positions of genomic elements, rather than active molecular interactions (*Rutledge et al., 2015*; *Tjong et al., 2012*).

Although this simplicity is attractive, diploidy and variable environmental conditions may add complexity to yeast genome conformation. In diploid yeast, homologous chromosomes can pair not only in meiosis, but also during mitotic growth (*Burgess and Kleckner, 1999*; *Burgess et al., 1999*; *Dekker et al., 2002*; *Weiner and Kleckner, 1994*), as they do in *Drosophila* (*Metz, 1916*). However, the extent of mitotic homolog pairing has been debated due to discrepancies between studies (*Barzel and Kupiec, 2008*). One explanation for these discrepancies is potential artifacts in the microscopy methods used to detect pairing. In fluorescence in situ hybridization (FISH), signal loss can lead to false inference of colocalization (*Lorenz et al., 2003*). It has also been suggested that tagging of genomic loci with repetitive arrays of GFP for live-cell imaging can directly cause pairing via GFP dimerization (*Mirkin et al., 2014*). Furthermore, the Rabl-like orientation alone can create the appearance of homolog pairing if not controlled for, by juxtaposing chromosomal loci at the same distance from centromeres, including homologous loci (*Lorenz et al., 2003*). Discrepancies in the extent of pairing between studies might be attributable to variation in pairing strength across the genome; however, mitotic homolog pairing has only been examined at a few loci.

In both haploid and diploid yeasts, genome conformation can also change in response to environmental conditions. Genes that respond to signals like galactose induction (*GAL1*, *HXK1*), inositol starvation (*INO1*), oxidative stress, and heat shock (*HSP104*) have been observed by microscopy to relocate to the nuclear periphery upon activation via interactions with nuclear pore complexes (*Ahmed et al., 2010*; *Brickner and Walter, 2004*; *Brickner et al., 2016*; *Casolari et al., 2004*; *Dieppois et al., 2006*; *Dultz et al., 2016*; *Taddei et al., 2006*). Nuclear pore interactions can mediate clustering of genes that share Gene Recruitment Sequences, including homologous alleles (*Brickner et al., 2015*, *2012*, *2016*; *Randise-Hinchliff et al., 2016*), and even impact the

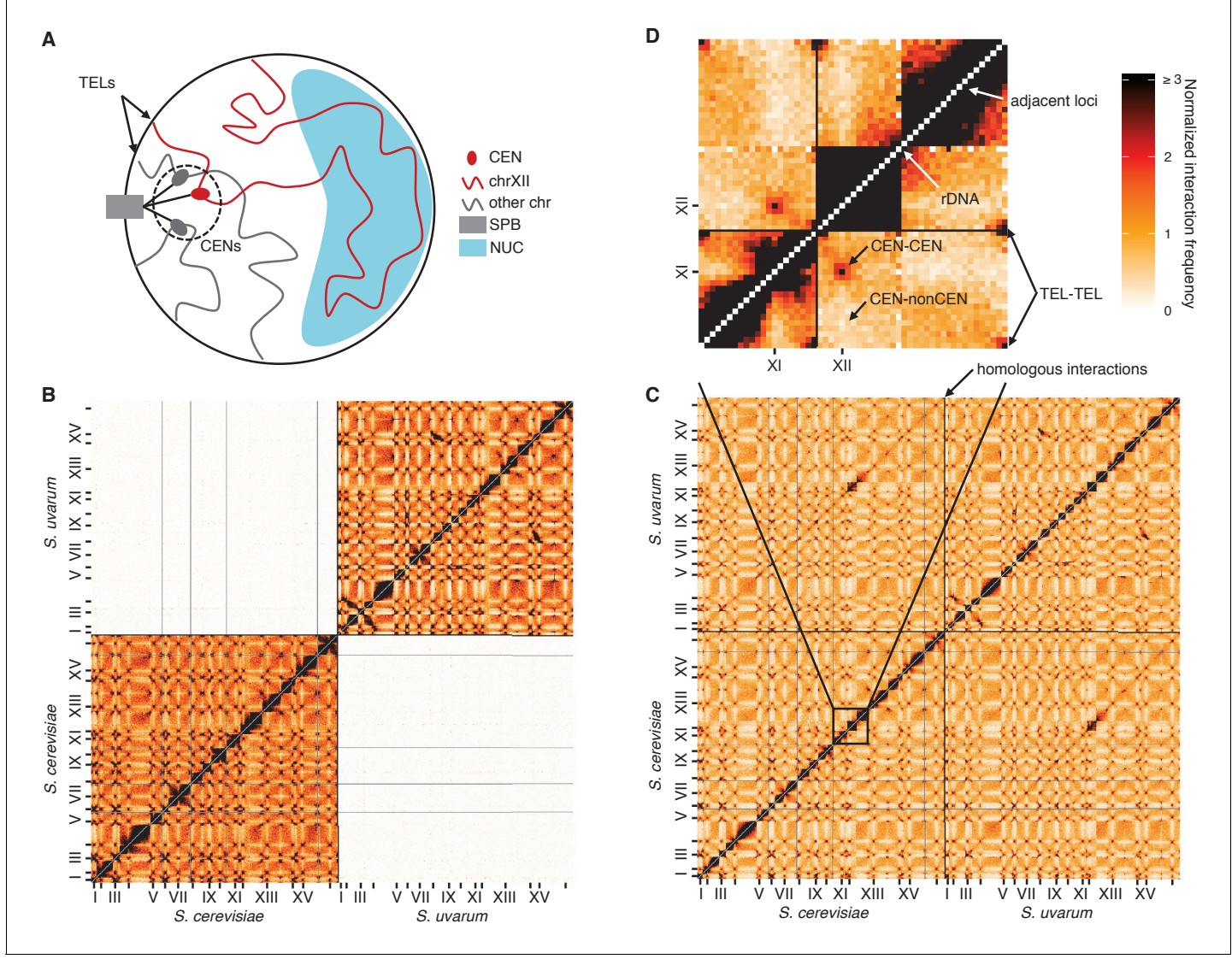

**Figure 1.** Diverged hybrids provide a genome-wide view of diploid chromosome conformation. (A) Schematic of the Rabl-like orientation. CEN, centromere; SPB, spindle pole body; TEL, telomere; NUC, nucleolus. (B) Hi-C contact map for saturated *S. cerevisiae* and *S. uvarum* mixture control, at 32 kb resolution. Each axis represents the *S. cerevisiae* genome followed by the *S. uvarum* genome in syntenic order, separated by a black line. Ticks indicate centromeres. Odd-numbered centromeres are labeled. Rows and columns with insufficient data are colored grey. (C) Hi-C contact map for saturated *S. cerevisiae* x *S. uvarum* hybrid, as in (B). (D) The portion of the map outlined in black in (C), is enlarged with annotated features of the Rabl-like orientation.

The following figure supplements are available for figure 1:

**Figure supplement 1.** Mappability of hybrid yeast genomes.

**Figure supplement 2.** Mixture control experiments for *S. cerevisiae* x *S. paradoxus* and *S. cerevisiae* x *S. cerevisiae* hybrids.

**Figure supplement 3.** Reproducibility of Hi-C across replicates and restriction enzymes.

**Figure supplement 4.** Revisions to *S. paradoxus* and *S. uvarum* reference genomes.

conformation of chromosomes well beyond the induced gene (*Dultz et al., 2016*). However, because such changes in conformation are primarily detected by microscopy, systematic studies of how inducible gene relocalization impacts global genome conformation have been lacking.

Here, we present a genome-wide analysis of diploid chromosome conformation in budding yeasts in multiple environmental conditions. We utilize hybrid yeasts resulting from mating diverged yeast species to perform homolog-resolved Hi-C. Our genomic approach allows us to more fully account for the Rabl-like orientation in assessing the extent of homolog pairing, and to detect whether some regions of the genome pair more strongly than others. We find that the strength of pairing varies across both growth conditions and the genome. Notably, the homologous *HAS1-TDA1* alleles on chromosome XIII pair specifically in galactose induction and saturated growth, but not during exponential growth in glucose. The condition-specific pairing is accompanied by relocalization to the nuclear periphery and in galactose requires Nup2, a component of the nuclear pore, suggesting a role for the nuclear pore complex. However, the genetic requirements of *HAS1-TDA1* relocalization and pairing differ from that of previously known relocalized genes, suggesting a potentially novel mechanism. Together, our results demonstrate the underappreciated complexity of the 3D organization of the yeast genome.

## Results

### Hi-C in hybrid yeasts provides a global view of diploid chromosome conformation

We performed Hi-C on interspecific hybrids between diverged *Saccharomyces* species to obtain the first genome-wide view of chromosome conformation in diploid yeasts. The sequence identity of homologous chromosomes in diploid *S. cerevisiae* precludes observation of interactions between them using sequencing-based methods. However, divergent *Saccharomyces* species can form stable hybrids (*González et al., 2006*; *Mertens et al., 2015*), e.g. between *S. cerevisiae* and *S. paradoxus* (90% nucleotide identity in coding regions [*Kellis et al., 2003*]) or its more distant relative *S. uvarum* (also known as *S. bayanus* var. *uvarum*; 80% nucleotide identity in coding regions [*Kellis et al., 2003*]). These interspecific hybrids are sufficiently diverged to allow straightforward sequence-level discrimination of homologs (*Figure 1—figure supplement 1A*) but have maintained nearly complete synteny (*Fischer et al., 2000*). For comparison, we also analyzed hybrids between *S. cerevisiae* strains Y12 and DBVPG6044, which are much less diverged (~99% nucleotide identity) (*Liti et al., 2009*). We confirmed the minimal impact of mapping and experimental artifacts by mapping Hi-C data from each individual species or strain (*Figure 1—figure supplement 1B–D*) and mixtures thereof (*Figure 1B*, *Figure 1—figure supplement 2*) to the hybrid reference genomes.

The most prominent features of Hi-C data from diploid yeast are the signatures of a Rabl-like orientation (*Figure 1C,D*). As in all Hi-C datasets, the contact map exhibits a strong diagonal signal indicating frequent intrachromosomal interactions between adjacent loci. In addition, pericentromeric regions interact frequently with one another, but infrequently with regions far from centromeres, as expected from the clustering of centromeres at the spindle pole body. Telomeric regions also preferentially interact, consistent with their clustering at the nuclear periphery. Finally, the rDNA-carrying chromosomes each behave as two separate chromosomes divided by the nucleolus, with frequent interactions on either side of the rDNA array but not across it.

### Homolog proximity exceeds the effects of the Rabl-like orientation

In addition to these previously described phenomena, we observed an off-diagonal line of increased interaction suggestive of homolog pairing (*Figure 1C*). Homologous loci tend to be closer together than nonhomologous loci in multiple assays, including microscopy (*Burgess et al., 1999*; *Weiner and Kleckner, 1994*), recombination efficiency (*Burgess and Kleckner, 1999*), and chromatin conformation capture (*Dekker et al., 2002*). Mitotic homolog pairing could be the result of transient pairing between homologous nucleosome-free DNA (*Danilowicz et al., 2009*; *Gladyshev and Kleckner, 2014*) or interactions among proteins bound to DNA (*Mirkin et al., 2014*). However, it has also been suggested that the observation of homolog proximity is an artifact of the Rabl-like orientation or microscopy methods (*Lorenz et al., 2003*; *Mirkin et al., 2014*). This debate remains

unresolved in part due to the targeted nature of previous studies, wherein each pair of homologous loci is only compared to a limited number of nonhomologous loci.

To systematically investigate whether homolog proximity can be explained by the Rabl-like orientation, we compared our experimental data from *S. cerevisiae* x *S. uvarum* hybrids to simulated data from a volume-exclusion polymer model of the Rabl-like orientation. This model did not explicitly encode homolog pairing (*Tjong et al., 2012*) and served as a negative control. We quantified homolog proximity by comparing the frequency of each interaction between a pair of homologous loci to the set of nonhomologous interactions involving either locus (*Figure 2—figure supplement 1*). This naive comparison appears to suggest strong homolog proximity in both experiments (*Figure 2A*, left panel), but in fact, the equally strong signal from the polymer model suggests that the apparent signal is a consequence of the Rabl-like orientation. We therefore controlled for the Rabl-like orientation by restricting comparisons to interactions with loci at a similar distance from the centromere (at a resolution of 32 kb), as previous studies have done (*Burgess et al., 1999*; *Lorenz et al., 2003*). Using this approach, we find that the polymer simulation still predicts strong homolog proximity (*Figure 2A*, middle panel), suggesting that the long-used approach of comparing homologous interactions to nonhomologous interactions at the same centromeric distance may not fully account for the Rabl-like orientation. Polymer models of the Rabl-like orientation suggest that short chromosomes interact preferentially, due to their dual telomeric tethering at the nuclear periphery and centromeric tethering at the spindle pole (*Tjong et al., 2012*). Therefore, we further restricted comparisons to loci on chromosome arms of similar length (within 25%). This additional restriction dramatically reduced the signal of homolog proximity for the polymer model, but not for the experimental data (*Figure 2A*, right panel).

Comparing homolog proximity across the genome, we noticed extensive interactions between the homologous chromosomes carrying the rDNA arrays (*Figure 2B*). To test whether this enrichment for interactions is due to sequence-dependent homolog pairing, we generated a translocation that swapped most of the centromeric half of *S. cerevisiae* chromosome XII with an equivalently sized portion of *S. cerevisiae* chromosome V, thereby moving the rDNA array to *S. cerevisiae* chromosome V. In this translocation-bearing strain, interactions between *S. uvarum* chromosome XII and *S. cerevisiae* chromosome V are enriched instead of *S. cerevisiae* chromosome XII (*Figure 2C,D* and *Figure 2—figure supplement 2A,B*), suggesting that homolog proximity of chromosomes carrying the rDNA arrays is due to the presence of the rDNA rather than the particular sequence of the chromosome that carries it. We propose that the rDNA-carrying chromosomes are uniquely positioned within the nucleus due to their tethering at the nucleolus (*Duan et al., 2010*) (*Figure 2—figure supplement 2C*). This shared tethering would then cause enhanced interactions between the homologous proximal and distal segments of these chromosomes and inflate the signal for apparent homology-dependent pairing.

Based on these findings, we excluded the rDNA-carrying chromosomes from estimates of homolog proximity. Even with these stringent constraints, we find that the observed interaction between homologous alleles exceeds that predicted based on the Rabl-like orientation (*Figure 2E*). Of note, the left arm of chromosome III and the right arm of chromosome IX exhibit particularly strong homolog proximity (*Figure 2B*); proximity at chromosome III is possibly due to pairing of the silenced mating-type loci (*Miele et al., 2009*).

In all hybrids, homolog proximity is substantially greater in saturated cultures approaching stationary phase than in exponential growth (*Figure 2E*), consistent with previous observations (*Burgess et al., 1999*). One explanation for this result is differences in the strength of sequence-dependent homolog pairing between growth conditions, perhaps mediated by differences in nucleosome positioning and DNA-bound proteins. However, this difference could also be a consequence of the reduced cell cycling coupled with loss of homolog proximity during S-phase (*Burgess et al., 1999*) or smaller nuclear size in cells approaching stationary phase (*Guidi et al., 2015*). To test whether we also observe cell cycle dependence of homolog proximity, we performed Hi-C on nocodazole-arrested cells, which were previously reported to exhibit reduced homolog proximity (*Burgess et al., 1999*). We find that nocodazole arrest does not substantially reduce homolog proximity in the diverged hybrid *S. cerevisiae* x *S. uvarum* (*Figure 2E*). However, it remains possible that the lack of S-phase cells in saturated cultures contributes to the difference in homolog proximity between exponentially growing and saturated cultures.

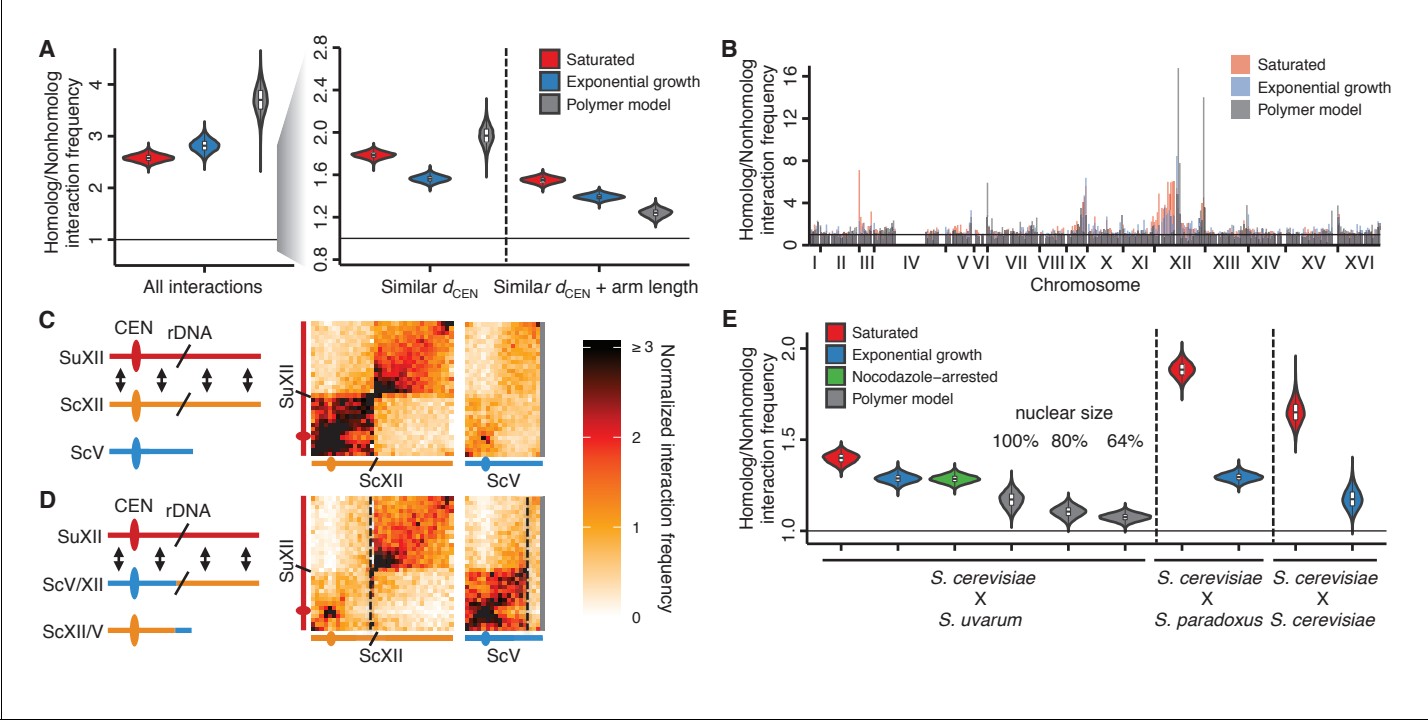

**Figure 2.** Homolog proximity exceeds predicted effects of Rabl-like orientation. (A) Violin plot of the distribution of 10,000 sampled estimates of genomic homolog proximity (ratio of homologous to nonhomologous interaction frequencies) in the *S. cerevisiae* x *S. uvarum* hybrid, as a function of increasing comparison stringency (left to right) to account for Rabl-like orientation (*Figure 2—figure supplement 1*). Saturated culture data are shown in red, exponential growth in blue, and simulated data from a homology-agnostic polymer model in grey. $d_{CEN}$, distance from centromere. Boxplot indicates median and interquartile range. Whiskers correspond to the highest and lowest points within 1.5× interquartile range. (B) Variation in homolog proximity across the *S. cerevisiae* x *S. uvarum* hybrid genome at 32 kb resolution, in saturated culture (red), exponential growth (blue), and the polymer model (grey). Nonhomologous interactions were restricted to similar centromere distance and chromosome arm length. Bins with insufficient data (<2 comparisons) are left blank. Data are plotted by *S. cerevisiae* genome position. x ticks indicate ends of chromosomes. (C and D) Schematics and Hi-C contact maps (at 32 kb resolution) of interactions between *S. uvarum* chromosome XII (SuXII) and either *S. cerevisiae* chromosome XII (ScXII) or *S. cerevisiae* chromosome V (ScV), in wild-type *S. cerevisiae* x *S. uvarum* hybrids (C) and a strain with a translocation between ScXII and ScV (D), both in saturated cultures. Exponential growth data are shown in *Figure 2—figure supplement 2*. Ovals indicate centromeres and slanted lines indicate the locations of rDNA arrays. Double-headed arrows indicate enhanced interactions. Dashed lines indicate translocation breakpoints. (E) Violin plot of homolog proximity across conditions, polymer models, and hybrids. *S. cerevisiae* x *S. cerevisiae* indicates hybrid between Y12 and DBVPG6044 strains. Calculated as in (A), but excluding chromosome XII and all 32 kb bins that include centromeres. Saturated culture data are shown in red, exponential growth in blue, nocodazole-arrested in green, and polymer models in grey. Boxplot indicates median and interquartile range. Whiskers correspond to the highest and lowest points within the 1.5× interquartile range.

The following figure supplements are available for figure 2:

**Figure supplement 1.** Schematic of homolog proximity analysis.

**Figure supplement 2.** rDNA-carrying chromosomes interact preferentially due to shared tethering.

We next sought to evaluate whether changes in nuclear size across growth conditions could explain the observed variation in homolog proximity. The nucleus is known to decrease in size in saturated cultures (*Guidi et al., 2015*), so we created alternate versions of the polymer model of the Rabl-like orientation with proportionally smaller nuclei, at 80% and 64% of the original size. In these models, smaller nuclear size led to decreased, rather than increased, homolog proximity (*Figure 2E*). These models suggest that the difference in homolog proximity between saturated and exponentially growing cultures cannot be explained by the effect of differences in nuclear size, and provide additional support for homolog pairing beyond the Rabl-like orientation.

## Relocalization of *GAL1* upon galactose induction alters genome conformation

We also searched our dataset for evidence of highly specific changes in genome conformation at the scale of individual genes. Microscopy studies have revealed inducible genes that relocate to the nuclear periphery upon activation due to association with nuclear pores, for example *GAL1* (*Brickner et al., 2016*; *Casolari et al., 2004*; *Dultz et al., 2016*), *INO1* (*Brickner and Walter, 2004*), *HXK1* (*Taddei et al., 2006*), *TSA2* (*Ahmed et al., 2010*), and *HSP104* (*Dieppois et al., 2006*), which can increase gene expression (*Ahmed et al., 2010*; *Brickner and Walter, 2004*; *Brickner et al., 2016*; *Taddei et al., 2006*). Although DNA interactions with components of the nuclear pore complex have been identified genome-wide by chromatin immunoprecipitation (*Casolari et al., 2004*), it remains unclear whether relocalization of specific genes impacts global genome conformation.

We first focused on the galactose metabolism gene *GAL1*. This gene and its neighbors *GAL7* and *GAL10* move upon galactose induction from their location near the spindle pole body to a nuclear pore complex at the nuclear periphery (*Casolari et al., 2004*; *Dultz et al., 2016*) (*Figure 3A*). Consistent with this expectation, we found using Hi-C that both *GAL1* loci interacted less with pericentromeric regions upon galactose induction (*Figure 3B–D*). Despite previous reports that the homologous *GAL1* loci preferentially interact with each other during galactose induction (*Brickner et al., 2016*; *Zhang and Bai, 2016*), we do not see a clear signal for increased pairing (*Figure 3—figure supplement 1*), perhaps because of the high basal interaction frequency between pericentromeric loci or the divergence between *S. cerevisiae* and *S. uvarum*.

## Novel inducible pairing of homologous *HAS1-TDA1* loci

Having established that we could detect the known inducible relocalization of the *GAL1* gene, we looked for other specific changes in genome conformation in the well-studied environmental conditions of galactose induction and growth saturation (approaching stationary phase). Surprisingly, we observed markedly increased interactions between homologous loci surrounding the genes *HAS1* and *TDA1* (subsequently abbreviated as '*HAS1-TDA1* loci') on chromosome XIII under both growth saturation and galactose induction, compared to standard exponential growth in glucose (*Figure 4A*). In fact, under inducing conditions, this interaction is among the strongest genome-wide, excluding pericentromeric and subtelomeric regions (top interaction out of over 83,000; *Figure 4—figure supplement 1*). No canonical galactose-induced genes are in or near this region. Nevertheless, this inducible homolog proximity appears to be evolutionarily conserved, as it occurs in all three tested interspecific hybrids, at least in saturated culture (*Figure 4A–C*; galactose not tested in all hybrids).

To explore whether this pairing depends on the presence of specific sequences, we created various deletions of the *S. cerevisiae* copy of the region, ranging from a 20 kb region from *NGL2* through *YMR295C* (*Figure 4—figure supplement 1A*) to a single 1 kb intergenic region containing the promoters for *HAS1* and *TDA1* (*HAS1pr-TDA1pr*; *Figure 4D*). Every deletion that included this intergenic region reduced the interaction frequency between *HAS1-TDA1* homologs in saturated growth conditions back to uninduced levels, indicating that this inducible pairing is sequence-dependent (*Figure 4E* and *Figure 4—figure supplement 2*). In contrast, deletion of the *HAS1* coding sequence had minimal impact, which shows that the deletion construct itself did not impede inducible pairing (*Figure 4E* and *Figure 4—figure supplement 2*). To test whether the *HAS1pr-TDA1pr* region is sufficient to produce inducible pairing, we moved the *S. cerevisiae* copy of this region to the left arm of *S. cerevisiae* chromosome XIV. The ectopic *HAS1pr-TDA1pr* allele exhibited inducible interactions with the *S. uvarum HAS1pr-TDA1pr*, although not to the same extent as the endogenous allele (*Figure 4—figure supplement 3*). The diminished extent of inducible pairing may reflect the contribution of chromosomal homolog pairing, which would be disrupted in the ectopic location, or of additional regions that are not sufficient to produce pairing on their own. To verify whether this pairing occurs in homozygous *S. cerevisiae* diploids in addition to diverged hybrids, we labeled both *HAS1-TDA1* loci with integrated LacO arrays targeted by LacI-GFP and measured the distance between them in a population of cells by confocal microscopy (*Figure 4F*). Consistent with our Hi-C data, the *HAS1-TDA1* homologs were closer together in galactose-induced and saturated cultures than in glucose (*Figure 4G,H* and *Figure 4—source data 1*).

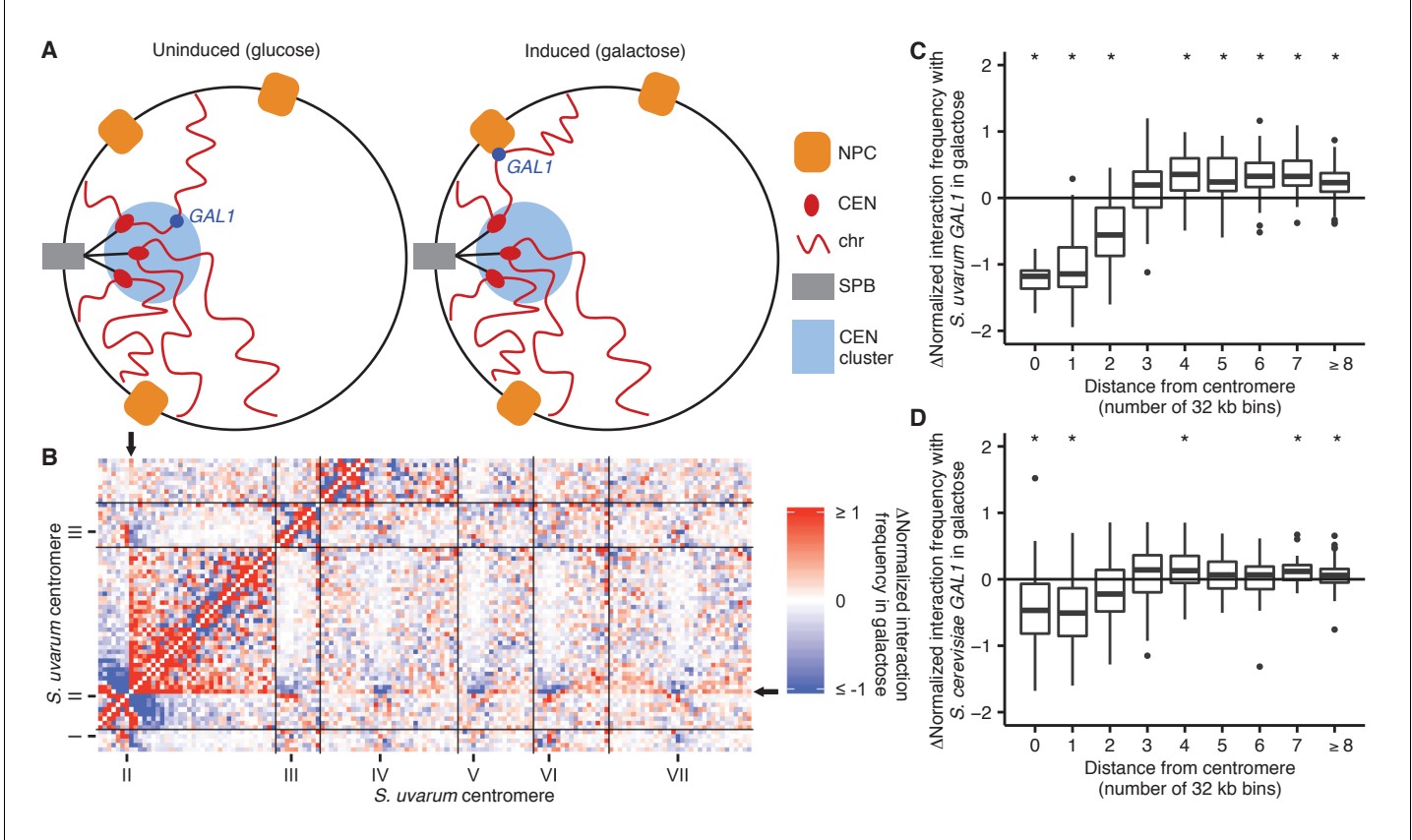

**Figure 3.** *GAL1* shifts away from centromeres upon galactose induction. (**A**) Schematic of *GAL1* positioning (dark blue) in glucose (left) and galactose (right). NPC, nuclear pore complex; CEN, centromere; chr, chromosome; SPB, spindle pole body. (**B**) Example region of differential Hi-C map of *S. cerevisiae* x *S. uvarum* hybrids in galactose vs. glucose, at 32 kb resolution. Interactions that strengthen in galactose are in red, while those that weaken are in blue. Ticks indicate centromeres; black lines indicate chromosomes. Arrows indicate location of *S. uvarum GAL1*. (**C**) Boxplot of the difference in *S. uvarum GAL1* interaction frequency in galactose vs. glucose across the *S. cerevisiae* x *S. uvarum* genome, excluding intrachromosomal interactions and binned by distance from the centromere (in 32 kb bins). Whiskers correspond to the highest and lowest points within the 1.5× interquartile range. *p<0.05 after Bonferroni correction (*n* = 9); Mann-Whitney test. Note: some outliers are beyond the plot range and are not shown. (**D**) Same as (**C**) for *S. cerevisiae GAL1*.

The following figure supplement is available for figure 3:

**Figure supplement 1.** *GAL1* homologs do not detectably pair during galactose induction.

## Nuclear pores play a role in *HAS1-TDA1* homolog pairing

Based on previous studies of relocalized genes (*Brickner et al., 2012*, *2016*), we hypothesized that pairing between the homologous *HAS1-TDA1* loci might be mediated by interactions of both alleles with nuclear pores (*Figure 5—figure supplement 1*). Therefore, we tested whether the *HAS1-TDA1* loci are relocalized to the nuclear periphery in a condition-dependent manner. We tagged the *HAS1-TDA1* locus with a LacO array as before and counted the proportion of cells in which *HAS1-TDA1* colocalized with the mCherry-labeled nuclear membrane, in haploid *S. cerevisiae*. Indeed, the *HAS1-TDA1* locus shifted to the nuclear periphery upon galactose induction and in saturated culture conditions (*Figure 5A* and *Figure 5—source data 1*). To confirm whether this inducible reorganization was dependent on association with nuclear pores, we repeated our analysis in strains with deletions of nuclear pore components *NUP2* or *NUP100*, or pore-associated protein *MLP2* (*Figure 5A* and *Figure 5—source data 1*). As in other cases of gene relocalization, Nup2 but not Nup100 was required for peripheral localization of the *HAS1-TDA1* locus. However, unlike other relocalized genes (*Ahmed et al., 2010*; *Brickner et al., 2016*; *Luthra et al., 2007*), *HAS1-TDA1* locus relocalization

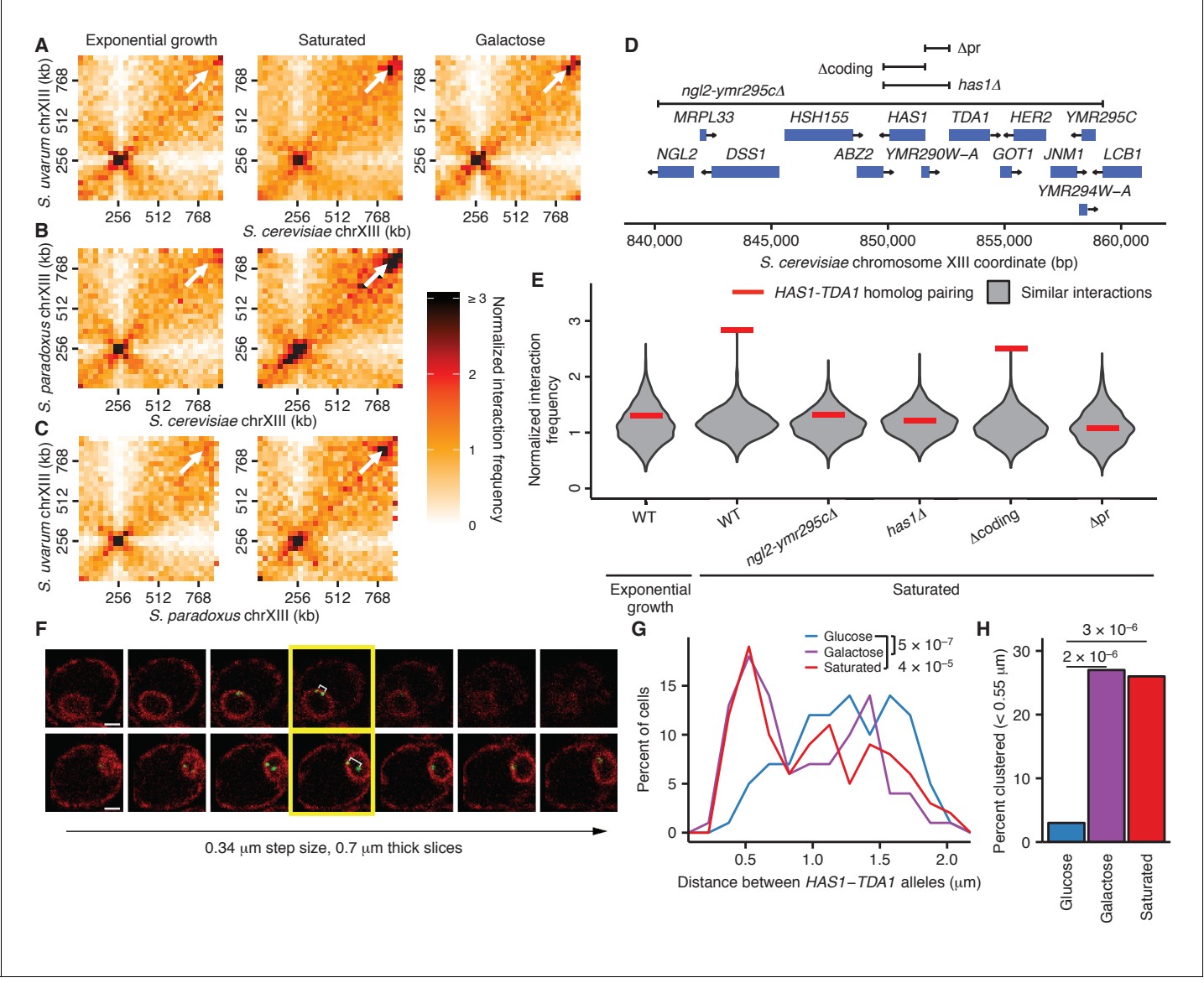

**Figure 4.** Inducible pairing of *HAS1-TDA1* homologs is evolutionarily conserved and sequence-specific. (A), (B), and (C) Hi-C contact maps of chromosome XIII interactions at 32 kb resolution in *S. cerevisiae* x *S. uvarum* (A), *S. cerevisiae* x *S. paradoxus* (B), and *S. paradoxus* x *S. uvarum* (C) hybrids in exponential growth (left column), saturated cultures (middle column), and in *S. cerevisiae* x *S. uvarum* hybrids (A), galactose (right column). White arrows indicate the interaction between the homologous *HAS1-TDA1* loci. (D) Genome browser shot of open-reading frames (ORFs; blue boxes) and tested deletions (brackets) in the *S. cerevisiae* region surrounding the genes *HAS1* and *TDA1*, from positions 840,000–860,000 (*Figure 4—figure supplement 1A*). Arrows indicate ends and directionalities of ORFs. (E) Strength of *HAS1-TDA1* homolog pairing at 32 kb resolution (red lines) compared to similar interactions (grey violin plots; i.e. interactions between an *S. cerevisiae* locus and an *S. uvarum* locus, where both loci are ≥15 bins from a centromere and ≥1 bin from a telomere, and not both on chromosome XII) in wild-type and deletion strains of *S. cerevisiae* x *S. uvarum*. See *Figure 4—figure supplement 2* for Hi-C contact maps of deletion strains. (F) Two example z-stacks of images used to measure distances between *HAS1-TDA1* alleles tagged with LacO arrays targeted by LacI-GFP (shown in green), with membranes labeled by ER04 mCherry (shown in red). The yellow outline indicates the images chosen for analysis. White brackets indicate the measured distance. Scale bar = 1 μm. (G) Distributions of the distance between the *HAS1-TDA1* alleles measured by microscopy in *S. cerevisiae* diploids in glucose (blue), galactose (purple), and saturated cultures (red). *n* = 100 for each condition. p-Values were calculated using the Wilcoxon rank sum test. (H) Frequency of *HAS1-TDA1* alleles less than 0.55 μm apart, measured as in (G), in glucose (blue), galactose (purple), or saturated cultures (red). p-Values were calculated using Fisher's exact test.

The following source data and figure supplements are available for figure 4:

**Source data 1.** Raw cell counts for *Figure 4G and H*.

**Figure supplement 1.** Exceptional inducible homolog pairing at *HAS1-TDA1* locus.

*Figure 4 continued on next page*

*Figure 4 continued*

**Figure supplement 2.** *HAS1-TDA1* homolog pairing does not shift nearby upon deletion.

**Figure supplement 3.** *HAS1-TDA1* homolog pairing is recapitulated ectopically by the *HAS1* and *TDA1* promoters.

did not require Mlp2, suggesting that the *HAS1-TDA1* locus may interact with nuclear pores via a distinct mechanism. We performed these initial analyses in haploids, to facilitate deletion of nuclear pore components, but pairing cannot occur in haploids. Thus, we confirmed that the *HAS1-TDA1* loci are peripherally relocalized in diploids, by reanalyzing the images we used to measure distances between *HAS1-TDA1* alleles in diploid cells (*Figure 5B* and *Figure 5—source data 2*). We then asked whether pairing of the *HAS1-TDA1* alleles only occurs at the periphery, by determining the proportion of cells in each category of peripheral localization that have paired *HAS1-TDA1* alleles (*Figure 5C* and *Figure 5—source data 2*). In fact, *HAS1-TDA1* alleles can remain paired in galactose in the nucleoplasm. However, this need not imply that the nuclear pores do not play a role in pairing. Previous studies of gene relocalization to nuclear pores have reported that the cell cycle affects when genes relocalize to the nuclear periphery; namely, genes tend to move to the nucleoplasm during S-phase (*Brickner and Brickner, 2010*). Alleles can remain paired in the nucleoplasm, but cannot actively pair during S-phase (*Brickner et al., 2012*). In agreement with these studies, the peripheral localization of *HAS1-TDA1* loci that we observe also exhibits cell cycle dependence (*Figure 5D,E* and *Figure 5—source data 2*), which may explain the presence of pairing in the nucleoplasm.

To test whether the nuclear pore complex is required for pairing as well as relocalization of the *HAS1-TDA1* loci, we performed Hi-C on an *S. cerevisiae* x *S. uvarum* hybrid strain with a homozygous deletion of *NUP2*. In this strain, *HAS1-TDA1* pairing was not observed in galactose, as expected, but still occurred at full strength in saturated growth (*Figure 5F*). These data indicate that Nup2 is required for *HAS1-TDA1* homolog pairing in galactose but not in saturated culture, suggesting distinct and/or additional mechanisms of pairing. To test biochemically whether the *HAS1-TDA1* loci interact with nuclear pores, we performed chromatin immunoprecipitation (ChIP) sequencing on the nuclear basket protein Nup60, which unlike the dynamic Nup2 cannot dissociate from the nuclear pore complex, tagged with the tandem affinity purification (TAP) tag (*Ghaemmaghami et al., 2003*), in haploid *S. cerevisiae* grown in either glucose or galactose. As expected, we observed a clear enrichment of the galactose metabolism gene cluster *GAL1-GAL10-GAL7* in the immunoprecipitated DNA from cells grown in galactose, compared to those grown in glucose (*Figure 5G*). In contrast, we observed little if any such enrichment of the *HAS1-TDA1* locus (*Figure 5H*). We also performed qPCR on the same ChIP DNA, which gave the same results (*Figure 5I* and *Figure 5—figure supplement 2*). Together, these data suggest that although *HAS1-TDA1* locus relocalization and pairing requires Nup2 in galactose, *HAS1-TDA1* differs from *GAL1* in how it interacts with the nuclear pore complex and may pair via alternative mechanisms as well, particularly in saturated growth.

## Transcription at the *HAS1-TDA1* locus

Nuclear pore association is thought to play a role in transcriptional regulation, generally but not always leading to greater or faster activation (*Akhtar and Gasser, 2007*; *Taddei, 2007*; *Taddei et al., 2010*). To explore how nuclear pore association might affect transcription at the *HAS1-TDA1* locus, we performed RNA sequencing on haploid *S. cerevisiae* grown in glucose, galactose, and saturated growth conditions. *HAS1* is downregulated in both pairing conditions (*Figure 6A*), whereas *TDA1* is weakly upregulated in both conditions (*Figure 6B*). However, transcriptional changes at *HAS1* and *TDA1* are relatively unremarkable; in both galactose and saturated growth, dozens to hundreds of genes are more strongly up- or downregulated than *HAS1* or *TDA1* (*Figure 6C,D*). For example, *GAL1* is upregulated nearly 1000-fold in galactose (*Figure 6C*). This suggests that nuclear pore association is not solely a function of strong transcriptional activation.

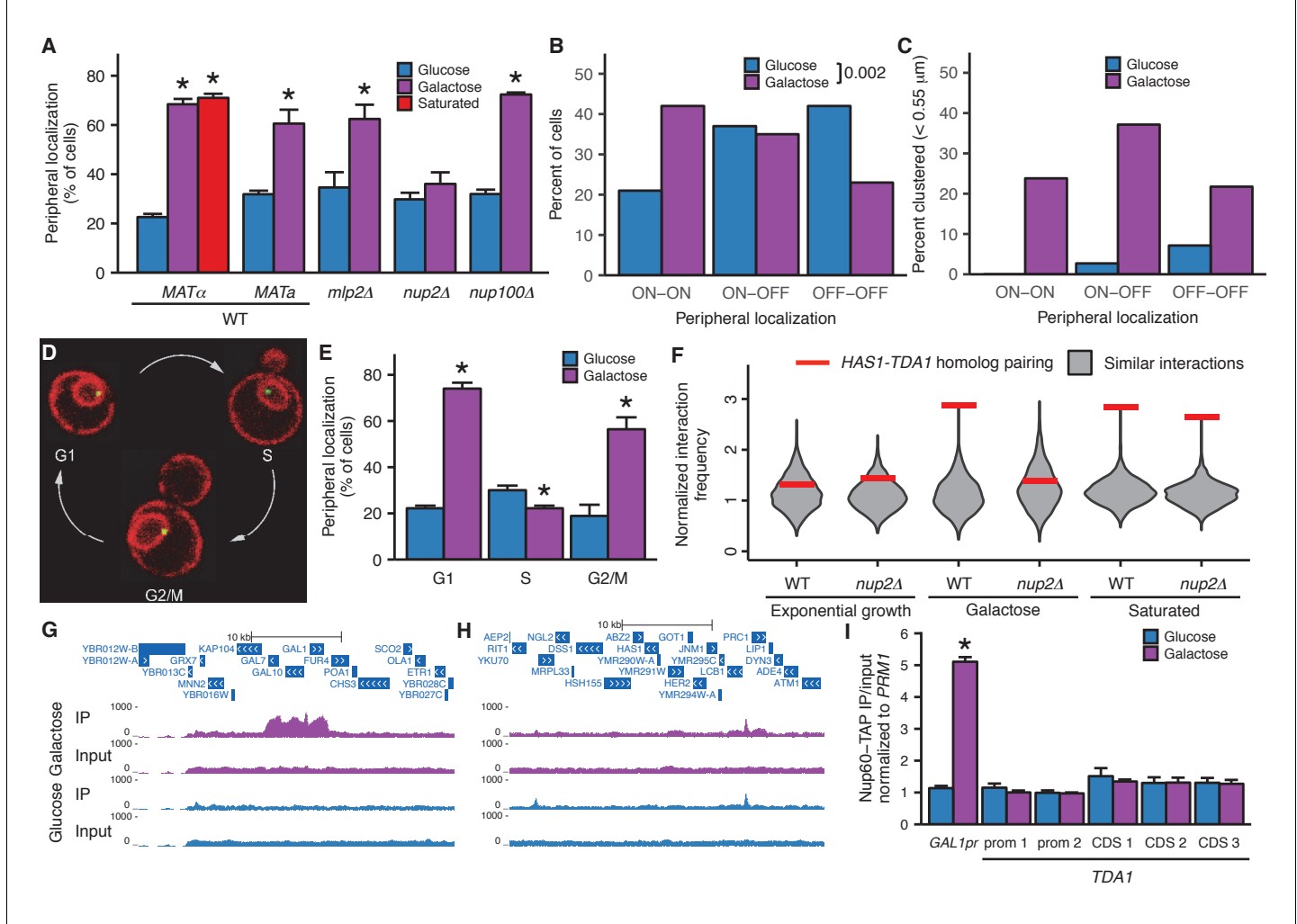

**Figure 5.** Inducible peripheral localization and pairing of *HAS1-TDA1* alleles involve nuclear pore interactions. (A) Proportions of haploid *S. cerevisiae* cells exhibiting peripheral *HAS1-TDA1* localization in strains with and without deletions of nuclear pore components, in glucose (in blue), galactose (in purple), or saturated culture (in red). Experiments were performed in biological triplicate, with $n \geq 30$ per experiment. *$p < 0.05$, Student's *t*-test. Center values and error bars represent mean ± s.e.m. (B) Proportion of diploid *S. cerevisiae* cells exhibiting two (ON-ON), one (ON-OFF), or zero *HAS1-TDA1* alleles with peripheral localization, in glucose (blue) and galactose (purple). p-Value calculated using chi-squared test. (C) Proportion of diploid *S. cerevisiae* cells with *HAS1-TDA1* alleles clustered (<0.55 μm apart), in glucose (blue) and galactose (purple) as a function of the peripheral localization of *HAS1-TDA1* alleles. Same images used as in (B) and *Figure 4G,H*. (D) Example images of cells in G1, S, and G2/M phases of cell cycle. (E) Proportions of haploid *S. cerevisiae* cells exhibiting peripheral *HAS1-TDA1* localization in different phases of the cell cycle, in glucose (in blue) and galactose (in purple). *$p < 0.05$, Student's *t*-test. Center values and error bars represent mean ± s.e.m. (F) Strength of *HAS1-TDA1* homolog pairing at 32 kb resolution (red lines) compared to similar interactions (grey violin plots; that is, interactions between an *S. cerevisiae* locus and an *S. uvarum* locus, where both loci are $\geq 15$ bins from a centromere and $\geq 1$ bin from a telomere, and not both on chromosome XII) in wild-type and homozygous nup2Δ strains of *S. cerevisiae* x *S. uvarum*. (G and H) Nup60-TAP ChIP-seq read coverage tracks for IP and input in galactose (purple) and glucose (blue), zoomed into a 35 kb region surrounding *GAL1-GAL10-GAL7* on chromosome II (G), and a 35 kb region surrounding *HAS1* and *TDA1* on chromosome XIII (H). (I) Nup60-TAP ChIP qPCR as IP/input normalized to the negative control *PRM1*, for the positive control *GAL1pr* and five sets of primers in *TDA1*. Primer sequences are provided in *Supplementary file 2*. Center values and error bars represent mean ± s.e.m. of three biological replicates. *$p < 0.05$, Student's *t*-test.

The following source data and figure supplements are available for figure 5:

**Source data 1.** Raw cell counts for *Figure 5A*.

**Source data 2.** Raw cell counts for *Figure 5B and C*.

**Source data 3.** Raw cell counts for *Figure 5E*.

*Figure 5 continued*

**Figure supplement 1.** Schematic of how nuclear pore association mediates homologous *HAS1* pairing.

**Figure supplement 2.** Mock-IP on Nup60-TAP.

## Discussion

Homologous chromosomes pair along their lengths leading up to and during meiosis, but may recognize each other and preferentially interact even in normal mitotic growth, perhaps to facilitate homology-directed repair or prepare for meiosis under stressful conditions. However, whether and to what extent this mitotic homolog pairing occurs has remained controversial, in part due to the lack of genome-wide data and the apparent homolog pairing caused by the Rabl-like orientation. We performed Hi-C in diverged hybrid diploid yeast, which allowed us to resolve homologous chromosomes (*Figure 1*) and thus infer homolog pairing strength on a genome-wide basis (*Figure 2B*). After using a polymer model of the Rabl-like orientation to calibrate our estimates, we find that even in hybrid diploids with homologs diverged to less than 80% nucleotide identity, homologous chromosomes do interact preferentially during mitotic growth, albeit subtly (*Figure 2E*). It would be interesting to compare the strength of pairing across hybrids with varying levels of divergence; however, our homolog pairing analysis requires filtering and stratifying genomic regions and thus may not be directly comparable across different reference genomes. Our data do not necessarily imply end-to-end chromosome alignment as occurs in meiosis. Instead, our data indicate an increased frequency of contact between homologous loci above the expectation based on random collisions, with substantial variation across the genome. Nevertheless, that such distant homology is sufficient for at least some homolog interactions is perhaps remarkable, and may hint at the role of DNA-bound proteins, which are more conserved than DNA, in mediating the interactions. Homolog

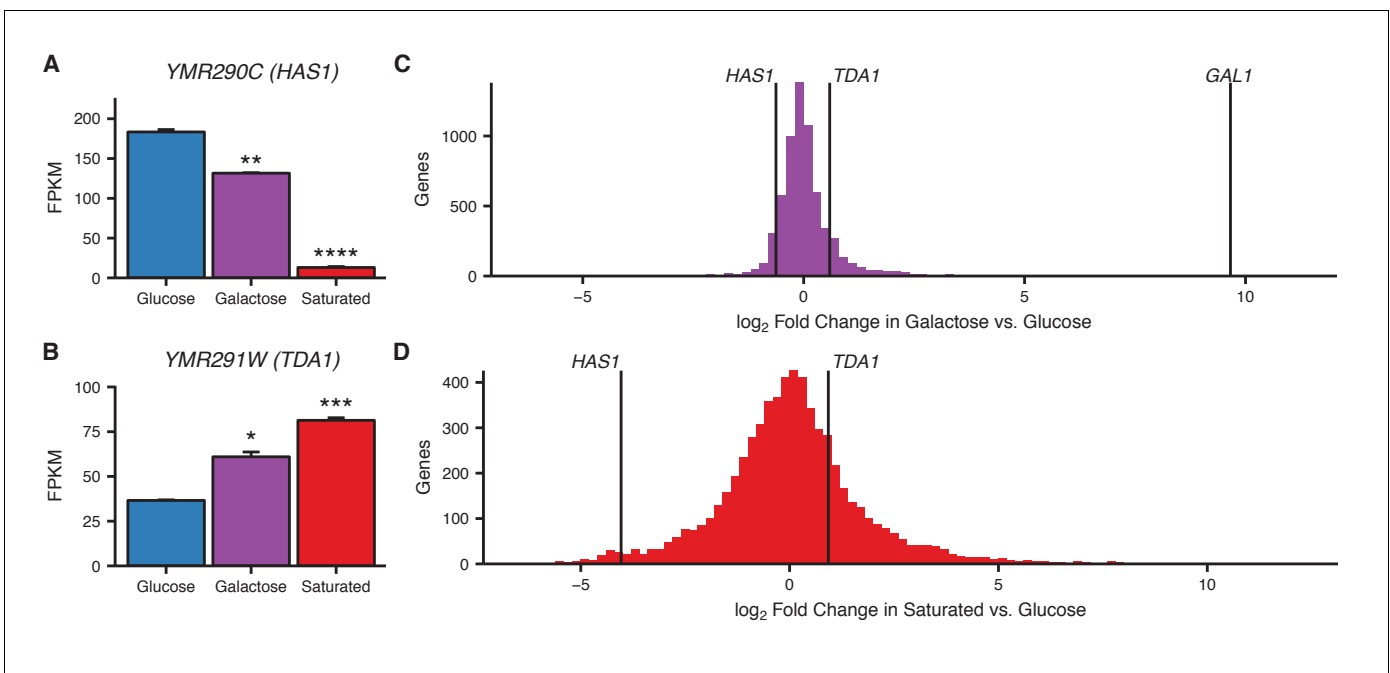

**Figure 6.** Transcriptional changes in galactose and saturated culture. (**A and B**) Bar plots of gene expression in haploid *S. cerevisiae* grown in glucose, galactose, or to saturation, for *HAS1* (**A**) and *TDA1* (**B**). Asterisks indicate p-values<0.05 (*), 0.01 (**), 0.001 (***), or 0.0001 (****), Student's *t*-test. Center values and error bars represent mean ± s.e.m. of three biological replicates (**C** and **D**) Histogram of log₂ fold change in gene expression in galactose (**C**) or saturated growth (**D**) compared to glucose. Vertical lines indicate values for *HAS1*, *TDA1*, and *GAL1* (**C** only).

pairing strength also depends on both growth conditions and genomic location, sometimes jointly: the homologous *HAS1-TDA1* loci on chromosome XIII pairs during saturated growth and galactose induction, but not exponential growth in glucose (*Figure 4*). This region is not remarkably conserved, suggesting that homolog pairing is at least partly due to specific interactions mediated by proteins, rather than direct DNA-DNA homology interactions (*Danilowicz et al., 2009*; *Gladyshev and Kleckner, 2014*).

In all tested hybrids, the *HAS1-TDA1* locus exhibits surprisingly strong homolog proximity (*Figure 4A–C*; *Figure 4—figure supplement 1B*). How does *HAS1-TDA1* pairing occur, and why? The nuclear pore component Nup2 seems to play a role, although not exclusively, in mediating pairing. The *HAS1-TDA1* locus moves to the nuclear periphery under pairing conditions, and both this relocalization and pairing are Nup2-dependent in galactose (*Figure 5A,F* and *Figure 5—source data 1*). However, Nup2 is not required for *HAS1-TDA1* pairing in saturated growth (*Figure 5F*). Together, the Nup2-independence of *HAS1-TDA1* pairing in saturated growth, the Mlp2-independence of *HAS1-TDA1* peripheral localization, and the lack of *HAS1-TDA1* enrichment in Nup60 ChIP suggest that *HAS1-TDA1* may interact with nuclear pores by a mechanism distinct from the previously studied *GAL1* and *INO1*, and possibly by different mechanisms in galactose and in saturated culture. However, more experiments are needed to fully elucidate the role and mechanism of the nuclear pore complex in *HAS1-TDA1* homolog pairing. Regardless of the molecular mechanism, nuclear pore interactions may confine the *HAS1-TDA1* alleles to the relatively small space near the nuclear periphery, thus speeding up the rate at which they randomly contact each other (*Figure 5—figure supplement 1*). Once in physical proximity, additional mechanisms such as protein-protein interactions between DNA-binding proteins could prolong the duration of contact, even after the alleles are no longer at a nuclear pore. Indeed, the presence of paired *HAS1-TDA1* alleles in the nucleoplasm suggests that such nuclear pore-independent pairing mechanisms may act at *HAS1-TDA1* (*Figure 5C* and *Figure 5—source data 2*). Interestingly, a recent study showed that Nup2 is involved in meiotic homolog pairing (*Chu et al., 2017*), suggesting that Nup2 may more generally play a role in homolog pairing.

Why do the homologous *HAS1-TDA1* alleles pair and relocalize to the periphery, and why does this interaction appear to be unique? Many genes associate with nuclear pores and relocalize to the nuclear periphery upon activation, including *GAL1*, but we do not observe strong pairing of *GAL1* homologs. It is possible that Hi-C may be failing to capture pairing at *GAL1* due to its pericentromeric location, but these data may also reflect particularly strong pairing at the *HAS1-TDA1* loci. Divergence between *S. cerevisiae* and *S. uvarum*, particularly in their galactose metabolism pathways (*Roop et al., 2016*), may also contribute to the lack of pairing at the *GAL1* locus. Lack of *GAL1* pairing need not correspond to lack of peripheral localization (*Brickner et al., 2016*); in our hybrid, the *S. cerevisiae* and *S. uvarum GAL1*-binding proteins may be able to each interact with the nuclear pores but not with each other.

Which gene is driving *HAS1-TDA1* homolog pairing, *HAS1* or *TDA1*, or both? Given the association between nuclear pore interactions and transcriptional activation, transcriptional changes in growth conditions that induce pairing may provide a clue. The genes at the *HAS1-TDA1* locus, *HAS1* and *TDA1*, demonstrate opposing changes in gene expression in galactose and saturated culture: *HAS1* is downregulated, whereas *TDA1* is upregulated. The upregulation of *TDA1*, although not particularly strong in magnitude (*Figure 6C,D*) may be of functional importance. Tda1 is a kinase required for phosphorylation of Hxk2, the primary hexokinase in yeast (*Kaps et al., 2015*; *Kettner et al., 2012*). Unphosphorylated Hxk2 can interact with Mig1 to repress various alternative carbon source metabolism genes in the presence of glucose. Phosphorylation of Hxk2 by Tda1 in low-glucose conditions prevents its interaction with Mig1 and thus leads to release from glucose repression. While we have not yet tested whether disrupting peripheral localization would affect *TDA1* or *HAS1* transcription, we hypothesize that nuclear pore interaction may aid the upregulation of *TDA1* in response to low-glucose concentrations, perhaps by facilitating efficient transcription or mRNA export from the nucleus.

Other questions remain about the mechanism and functional implications of *HAS1-TDA1* pairing and peripheral relocalization. Increased transcription may itself contribute to localization at nuclear pores via interactions between nascent mRNA and mRNA processing and export factors at nuclear pores (*Akhtar and Gasser, 2007*), and may be involved in establishment of *HAS1-TDA1* pairing as it is for *GAL1* (*Brickner et al., 2016*). However, given the abundance of other genes with similar or

greater changes in transcription that do not pair (*Figure 6*), transcription alone likely cannot explain our data. For some genes, nuclear pore interactions mediate rapid reactivation in a phenomenon termed epigenetic transcriptional memory (*D'Urso and Brickner, 2017*; *D'Urso et al., 2016*; *Light et al., 2010*); it is also possible that the nuclear pore interactions with *HAS1-TDA1* may be involved in epigenetic transcriptional memory. The pairing of the *HAS1-TDA1* alleles may also serve a distinct function, potentially including *trans* gene regulation like at *GAL1* (*Zhang and Bai, 2016*), but further experiments are needed to test this possibility.

The principles and functional implications of genome conformation remain open questions. Although the budding yeast *S. cerevisiae* is thought to have a simple genome organization, it serves as a versatile and relevant model system amenable to integrating multiple approaches to studying genome conformation, including Hi-C, polymer simulations, live-cell imaging, and genetic perturbations. While yeast nuclear organization may differ from that of other eukaryotes in some ways, our findings may, nevertheless, be applicable to other organisms: recent studies in the fruit fly *Drosophila* have provided evidence for the generality of the role of nuclear pores in transcriptional regulation first observed in yeast (*Pascual-Garcia et al., 2017*). Our study illustrates both the utility of combining orthogonal methodologies and that we have much more to learn about genome organization, even in the simple budding yeast.

## Materials and methods

### Strain construction

All yeast strains used in this study are listed in *Supplementary file 1*. All primers used in this study, including those used for generation and validation of strains, are listed in *Supplementary file 2*.

Hybrid strains were created by mating haploid strains and then performing auxotrophic selection.

The ScV-ScXII translocation, *S. cerevisiae* x *S. uvarum* strain was generated by first creating the translocation in the haploid *S. cerevisiae* strain BY4742, followed by mating with haploid *S. uvarum*. A cassette containing *hphMX* followed by the first half of *URA3*, an artificial intron, and a *lox71* site was amplified from pBAR3 (*Levy et al., 2015*) and integrated into the intergenic region between *YLR150W* and *YLR151C*. A second cassette containing a *lox66* site, an artificial intron, the second half of *URA3*, and *natMX* was amplified from pBAR2-natMX (pBAR2 [*Levy et al., 2015*] with *natMX* in place of *kanMX*) and integrated into the intergenic region between *YER151C* and *YER152C*. The translocation was induced by transforming the resulting strain with the galactose-inducible Cre plasmid pSH47-kanMX (pSH47 (*Güldener et al., 1996*) with *kanMX* in place of *URA3*), and then inducing Cre recombination by plating on YP + galactose medium. Successful translocation strains were selected by growing in medium lacking uracil, and verified by PCR across the translocation junctions. This *S. cerevisiae* strain was then mated with *S. uvarum* strain ILY376.

Heterozygous deletion strains were made in *S. cerevisiae* x *S. uvarum* hybrids, by replacing regions of interest with the *hphMX* cassette. Homozygous deletion strains were made by making deletions in haploids and then mating the haploid strains. Strains were verified by PCR across each deletion junction.

The knock-in strain was made by integrating the *HAS1pr-TDA1pr* region followed by the *natMX* cassette into the region between *YNL266W* and *YNL267W* (*PIK1*) on *S. cerevisiae* chromosome XIV in the *S. cerevisiae* x *S. uvarum* hybrid YMD3269 (*HAS1pr-TDA1pr* deletion).

Plasmids pAFS144 (*Straight et al., 1996*), p5LacIGFP (*Randise-Hinchliff et al., 2016*), pER04 (*Randise-Hinchliff et al., 2016*), and pFA6a-kanMX6 (*Longtine et al., 1998*) have been described. To tag *HAS1-TDA1* with the LacO array, 1 kb downstream of the *HAS1* ORF was PCR amplified and TOPO cloned to create pCR2.1-*HAS1_3'UTR*. Plasmid p6LacO128-*HAS1* was made by inserting *HAS1* from pCR2.1- *HAS1_3'UTR* into p6LacO128 (*Brickner and Walter, 2004*).

### Hi-C

Cells were grown overnight, shaking at 30°C (room temperature for *S. uvarum*) in YPD medium (1% yeast extract, 2% peptone, 2% dextrose), YP + raffinose (2%), or YP + galactose (2%). For saturated culture samples, they were crosslinked at this point by resuspension and incubation in 1% formaldehyde in PBS for 20 min at room temperature. Crosslinking was quenched by addition of 1% w/v solid glycine, followed by incubation for 20 min and a PBS wash. For all other experiments, fresh cultures

were inoculated to $OD_{600}$ = 0.1 in appropriate medium and grown to $OD_{600}$ = 0.6–0.8. Exponential growth samples were crosslinked at this point, while for nocodazole-arrested samples, cultures were supplemented with 15 μg/mL nocodazole and grown at 30°C for 2 hr following addition of drug prior to crosslinking. Arrested cultures were checked by flow cytometry. For mixture controls, samples were mixed prior to crosslinking.

Hi-C libraries were created as described (*Burton et al., 2014*) with the exceptions that the restriction endonuclease Sau3AI or HindIII was used to digest the chromatin and the Kapa Hyper Prep kit (Kapa Biosystems, Wilmington, MA) was used to create the Illumina library instead of the Illumina TruSeq kit. Libraries were pooled and sequenced on an Illumina NextSeq 500 (Illumina, San Diego, CA), with 2 × 80 bp reads for interspecific hybrids and 2 × 150 bp reads for intraspecific *S. cerevisiae* hybrids. Hi-C libraries were similar across the two restriction enzymes and biological replicates (*Figure 1—figure supplement 3*). All Hi-C libraries are listed in *Supplementary file 3*.

## Reference genomes

The *S. cerevisiae* references and annotations were downloaded from the *Saccharomyces* Genome Database (version R64.2.1). The *S. paradoxus* and *S. uvarum* references and annotations were downloaded from saccharomycessensustricto.org (*Scannell et al., 2011*) but modified to correct misassemblies evident based on synteny and Hi-C data (*Figure 1—figure supplement 4*). *S. paradoxus* chromosome IV was rearranged so bases 1–943,469 were followed by 1,029,253–1,193,028, then 1,027,718–1,029,252, then 943,470–1,027,717 in reverse order, followed by the remainder of the chromosome. *S. uvarum* chromosome III was rearranged so bases 219,500 onward were placed at the beginning (left end) of the chromosome, followed by the first 219,399 bases, and then new sequence determined by Sanger sequencing with primers CATTCCCATTTGTTGATTCCTG and GGA TTCTATTGTTGCTAAAGGC: TAATAAGGAAGAACTGCTTATTCTTAATTATTTCTACCTACTAAAC TAACTAATTATCAACAAATATCATCTATTTAATAGTATATCATCACATGCGGTGTAAGAGGATGACA TAAAGATTGAGAAACAGTCATCCAGTCTAATGGAAGCTCAAATGCAAGGGCTGATAATGTAA TAGGATAATGAATGACAACGTATAAAAGGAAAGAAGATAAAGCAATATTATTTTGTAGAATTA TCGATTCCCTTTTGTGGATCCCTATATCCTCGAGGAGAA. *S. uvarum* chromosomes X and XII were also swapped, based on homology to *S. cerevisiae*. The *S. cerevisiae* Y12 and DBVPG6044 strain references were sequenced to 145- and 315-fold coverage using the PacBio (Pacific Biosciences, Menlo Park, CA) single-molecule, real-time (SMRT) sequencing platform with P6-C2 chemistry. Each genome was assembled with FALCON (*Chin et al., 2016*), version June 30, 2015 hash: cee6a58, and polished with Quiver (*Chin et al., 2013*) version 1.1.0 to generate chromosome-length contigs (with the exclusion of chromosome XII, which was split at the rDNA array, and Y12 chromosome XIV, which was split into one large and one small contig). To call centromeres in *S. paradoxus*, we searched the region on each chromosome between the genes homologous to those nearest the centromeres in *S. cerevisiae* (e.g. *YEL001C* and *YER001W* on chromosome V) for the sequence motif $N_2TCAC(A/G)TGN_{95-100}CCGAAN_6$ (based on an alignment of *S. uvarum*, *S. mikatae*, and *S. kudriavzevii* centromeres [*Scannell et al., 2011*]) or its reverse complement. When this motif was absent (chromosomes VII and VIII), we called the centromere as the middle 120 bp of the region. To call centromeres in Y12 and DBVPG6044, we mapped the *S. cerevisiae* S288C centromere sequences to the new references.

## Theoretical mappability analysis

Simulated reads for each hybrid genome (as in experimental data, 80 bp for interspecific hybrids and 150 bp for the intraspecific *S. cerevisiae* hybrid) were generated by taking sequences of the read length at 10 bp intervals. These reads were then remapped to the hybrid genome using bowtie2 (*Langmead and Salzberg, 2012*) with the –very-sensitive parameter set. The proportion of reads that mapped with mapping quality $\geq$30 to the correct location was then calculated.

## Hi-C data analysis

Sequencing reads were first pre-processed using cutadapt (*Martin, 2011*): reads were quality-trimmed (option -q 20), trimmed of adapter sequences, and then trimmed up to the ligation junction (if present), excluding any read pairs in which either read was shorter than 20 bp after trimming (option -m 20). The two reads in each read pair were then mapped separately using bowtie2

(*Langmead and Salzberg, 2012*) with the –very-sensitive parameter set. For interspecific hybrids, reads were mapped to a combined reference containing both species references, where if secondary mappings were present the best alignment must have a score ≥10 greater than the next best alignment. For intraspecific *S. cerevisiae* hybrids, reads were mapped separately to both strain references, keeping only read pairs in which both reads mapped to both references—perfectly to one reference and with ≥2 mutations including ≥1 substitution to the other. PCR duplicates (with identical fragment start and end positions) were removed, as were read pairs mapping within 1 kb of each other or in the same restriction fragment, which represent either unligated or invalid ligation products. The genome was then binned into 32 kb fragments (except the last fragment of each chromosome), and the number of read pairs mapping to each 32 kb genomic bin was counted based on the position of the restriction sites that were ligated together. Due to gaps in the reference genomes of *S. uvarum*, some repetitive sequences were only represented once and therefore artifactually mapped uniquely; therefore, reads mapping to annotated repetitive sequences were masked from further analysis. Similarly, gaps in the *S. paradoxus* reference led to mismapping of reads to the corresponding *S. cerevisiae* sequence; therefore, for *S. cerevisiae* x *S. paradoxus* libraries we masked regions in the *S. cerevisiae* genome where >1 read from a *S. paradoxus* Hi-C library mapped, and vice versa. We took a similar approach to mask regions of the Y12 and DBVPG6044 references that were prone to mismapping, as estimated by haploid Y12 and DBVPG6044 Hi-C libraries. For knock-in experiments, the *HAS1pr-TDA1pr* region was masked to account for its altered genomic location. The resulting matrices were then normalized by excluding the diagonal (interactions within the same genomic bin), filtering out rows/columns with an average of less than one read per bin, and then multiplying each entry by the total number of read pairs divided by the column and row sums.

## Polymer model

The volume-exclusion polymer model of the Rabl-like orientation was a modified version of the Tjong et al. tethering model (*Tjong et al., 2012*). Briefly, beads representing segments of the genome are randomly positioned and then adjusted until constraints (e.g. consecutive beads must be adjacent, and no two beads can occupy the same space) are met. The model was extended from 16 chromosomes to 32, with the lengths of the *S. cerevisiae* and *S. uvarum* chromosomes. The parameters for nuclear size, centromeric constraint position and size, telomeric constraint at the nuclear periphery, and nucleolar position and size were scaled by a factor of 1.25 to reflect the roughly doubled volume of diploid nuclei (cell volume correlates with ploidy (*Mortimer, 1958*), and nuclear volume correlates with cell volume [*Jorgensen et al., 2007*]). To test the effect of smaller nuclei, all parameters were scaled by a factor of 0.8 or 0.64 from this initial diploid model. For each model, the modeling procedure was repeated 20,000 times to create a population of structures. From this population, we simulated Hi-C data by calling all beads within 45 nm of each other as contacting each other, and then counting the number of contacts between each pair of 32 kb bins. The resulting matrix of counts was normalized using the same pipeline as the experimental Hi-C data.

## Homolog proximity analysis

In order to assess homolog proximity genome-wide, we first determined which bins represented interactions between homologous sequences, and then compared the normalized interaction frequencies in those bins compared to a set of 'comparable' nonhomologous bins.

In the interspecific hybrids, we determined homology by counting the number of starts or ends of one-to-one homologous gene annotations falling into each bin. Genes whose 'SGD' and 'BLAST' gene annotations differed were ignored. To find homologous interaction bins for genomes 1 and 2, for each bin of genome 1 we considered the bin in genome 2 where the most homologous gene ends fell to be homologous.

In the intraspecific hybrids where inter-strain mapping was much more reliable, we simulated 150 bp reads from the Y12 genome at 10 bp intervals, then mapped them to the DBVPG6044 reference. Here, for each bin of genome 1 we considered the bin in genome 2 where the most reads mapped with MAPQ ≥30 to be homologous.

To eliminate minor 'homology' arising from repetitive sequences (e.g. telomeres), we excluded isolated homologous interaction bins lacking any other homologous interaction bins within two bins.

To fully exclude homologous interactions from our estimates of nonhomologous interactions, any interaction bins within 2 bins of homologous interaction bins were excluded from analyses.

After determining homologous bins, we compared each homologous bin to other intergenome interactions (i.e. between chromosomes from different species/strains) involving one of the two genomic bins involved in the homologous interaction. To control for the effects of the Rabl-like orientation, we further filtered the nonhomologous interaction bins for those in which the centromeric distance (in units of 32 kb bins) was equivalent, and then for those in which the chromosome arm lengths of the two loci were within 25% of each other (in units of 32 kb bins). We also considered exclusion of the rDNA carrying chromosome XIIs as well as the centromeric bins, for which we could not fully control chromosome arm lengths. In all cases, we only considered homologous bins with at least two comparable nonhomologous bins.

To estimate genome-wide homolog proximity, we compared the sum of normalized interaction frequencies across the homologous bins to those of an equal number of randomly chosen nonhomologous bins, one comparable to each homologous bin, with replacement. We repeated this 10,000 times to obtain a distribution of genomic homolog proximity.

To obtain a view of homolog proximity strength across the genome, we compared the normalized interaction frequency in each homologous bin to the median of that in the similar nonhomologous bins, and then plotted the ratio of homologous/nonhomologous across the *S. cerevisiae* genome.

## Confocal microscopy

Gene positioning at the nuclear periphery and inter-allelic clustering were determined as described previously (*Brickner and Walter, 2004*; *Brickner et al., 2016*; *Egecioglu et al., 2014*). Briefly, cells bearing an array of 128 Lac operators integrated downstream of the *HAS1* coding sequence and expressing both the ER04 mCherry membrane marker (*Egecioglu et al., 2014*) and the GFP-LacI (*Robinett et al., 1996*) were imaged on a Leica SP5 line-scanning confocal microscope (Leica Microsystems, Wetzlar, Germany).

Cultures were grown in synthetic minimal media with 2% glucose or 2% galactose overnight at 30°C with constant shaking and harvested in log phase ($OD_{600}$ <0.5) or late log/stationary phase ($OD_{600}$ >1.0). Unless noted, cultures were grown in the designated media overnight prior to imaging. Cultures were concentrated by brief centrifugation, and then 1 µl was spotted onto a microscope slide for visualization.

For all experiments, cells were illuminated at 10–15% power with 488 nm and 561 nm using argon and diode pumped solid state lasers, respectively. Stacked images of ~150 µm x 150 µm fields were collected; ~ 15–20 z-slices of 0.34 µm thickness each. The optical thickness of the slices is ~0.73 µm. The z-slice in which the green dot(s) is most focused and bright is selected for analysis (*Figure 4F*).

For peripheral localization experiments, cells in which the center of the dot colocalizes with the nuclear envelope, as measured by mCherry fluorescence, are scored as peripheral. All other cells are scored as nucleoplasmic. Cells in which the dot was at the top or bottom of the nucleus were excluded. Each experiment was performed three times, counting ~30 cells per replicate . The percent of cells scored as peripheral was averaged and the standard error of the mean was calculated. Student's *t*-test was used to compare these distributions.

To monitor inter-allelic clustering of the *HAS1-TDA1* locus, haploid strains bearing the LacO array integrated downstream of *HAS1* were mated to create a diploid strain . These strains were imaged as above and, in cells in which the two alleles were either in the same z-slice or adjacent z-slices , the distance between the centers of the dots was measured using LAS AF software. Cells in which the two dots were not in the same or adjacent z-slices, or cells in which the two dots were unresolvable, were excluded. For each experiment, 100 cells were measured and both the distribution of distances among 0.15 µm bins and the fraction of cells in which the two alleles were <0.55 µm was calculated. To compare distributions, the Wilcoxon Rank Sum test was used. To compare the fraction of the cells in which the two alleles were <0.55 µm, Fisher's exact test was used.

## Chromatin immunoprecipitation

Chromatin immunoprecipitation was performed as in (*Egecioglu et al., 2014*), with modifications. Nup60-TAP yeast (*Ghaemmaghami et al., 2003*) were grown overnight, diluted to $OD_{600}$ = 0.125– 0.15 in 50 ml medium, then grown to $OD_{600}$ = 0.75–0.85 at 30°C in either YPD (for glucose samples)

or YP +2% galactose (for galactose samples), then crosslinked with 1% formaldehyde (v/v) for 5 min at room temperature. Crosslinked cells were quenched in 150 mM glycine, washed twice in Tris-buffered saline, and then stored at −80°C.

Crosslinked cell pellets were resuspended in 500 μl lysis buffer with 1x cOmplete Protease Inhibitor tablet (Roche, Basel, Switzerland), and then 700 μl of 500 μm acid-washed glass beads were added. The cells were vortexed for 12 min total, in cycles of 2 min shaking and 2 min resting on ice. The lysate was pelleted and resuspended in fresh lysis buffer, and then sonicated for 3 × 10 min runs on a Diagenode Bioruptor (Diagenode, Liège, Belgium), on high power with cycles of 30 s on and 30 s off, to an average of ~300 bp. The sonicate was cleared by centrifugation, and the resulting supernatant was split into an input aliquot (1/20 of IP volume) and two halves for the IP and mock-IP (BSA instead of antibody). For each sample, 5 μg of anti-TAP antibody (Thermo Fisher Scientific, Waltham, MA; #CAB1001, Lot #RL240352) was used with 10 μl Dynabeads Protein A (Thermo). The antibody or BSA was incubated with pre-washed beads for 2 hr and then washed twice in fresh lysis buffer before being added to the lysate and then incubated overnight. After washing the beads four times in lysis buffer and eluting, the eluate was reverse crosslinked and then treated with RNase A and Proteinase K. DNA was purified using Zymo ChIP DNA Clean & Concentrator (Zymo Research, Irvine, CA) and eluted in 30 μl water.

IP/mock-IP samples were diluted 1:8 and inputs were diluted 1:320, and then 4 μl were used in each 10 μl qPCR reaction. qPCRs were performed in triplicate in 384-well plates on a ViiA7 (Applied Biosystems, Foster City, CA), with Kapa Robust 2G Hot Start 2x master mix (thermocycling as recommended, with 20 s extension for 40 cycles) and 0.2x SYBR Green I dye. CT values were calculated and normalized to a genomic DNA standard curve using the ViiA7 software. IP/input ratios were normalized to those for the negative control *PRM1*.

ChIP-seq libraries were prepared using the Accel-NGS 2S Plus DNA Library Kit (Swift Biosciences, Ann Arbor, MI), from equal volume pools of the biological replicates, either 1 ng total from input samples or 21 μl of total IP sample. Input and IP libraries were amplified for six and seven cycles, respectively. Libraries were sequenced to ~5–6 million read pairs using 2 × 75 bp reads on an Illumina MiSeq.

## ChIP-seq analysis

Sequencing reads were first pre-processed using cutadapt (*Martin, 2011*): reads were quality-trimmed (option -q 20), trimmed of adapter sequences, excluding any read pairs in which either read was shorter than 28 bp after trimming (option -m 28). The two reads in each read pair were then mapped jointly to the sacCer3 *S. cerevisiae* reference using bowtie2 (*Langmead and Salzberg, 2012*) with the –very-sensitive parameter set, requiring the two reads to be within 2000 bp of each other (option -X 2000). Fragments (read pairs) in which both reads had a mapping quality score of at least 30 were then deduplicated by fragment start and end positions and then aggregated into a coverage track using bedtools (*Quinlan and Hall, 2010*). Genome browser tracks were generated using the UCSC Genome Browser (*Kent et al., 2002*).

## RNA sequencing

BY4741 yeast were grown overnight in YPD (for exponential growth and saturated samples) or YP +2% galactose (for galactose samples) at 30°C, then diluted to $OD_{600}$ = 0.1–0.125 in 50 ml medium, and then grown to $OD_{600}$ = 0.5–0.6, pelleted and stored at −80°C. RNA was purified using acid phenol extraction, and then treated with the DNA-*free* DNase kit (Thermo). Illumina libraries were then prepared using the TruSeq RNA Library Prep Kit v2 (Illumina), with six cycles of amplification. Libraries were sequenced to ~15–20 million read pairs using 2 × 75 bp reads on an Illumina NextSeq 500.

## RNA-seq analysis

Sequencing reads were first pre-processed using cutadapt (*Martin, 2011*): reads were quality-trimmed (option -q 20), trimmed of adapter sequences, excluding any read pairs in which either read was shorter than 28 bp after trimming (option -m 28). The two reads in each read pair were then mapped jointly to the sacCer3 *S. cerevisiae* reference using bowtie2 (*Langmead and Salzberg, 2012*) with the –very-sensitive parameter set, requiring the two reads to be within 500 bp of each

other (option -X 500). Fragments (read pairs) in which both reads had a mapping quality score of at least 30 were overlapped with annotated genes using HTSeq (*Anders et al., 2015*). Global fold-change analyses were performed using DESeq2 (*Love et al., 2014*).

## Code availability

Code for all bioinformatic analyses is available at https://github.com/shendurelab/HybridYeastHiC (*Kim, 2017*). A copy is archived at https://github.com/elifesciences-publications/HybridYeastHiC.

## Data availability

GEO accession number: GSE88952

## Acknowledgements

We thank G Yardimci, G Bonora, M Kircher, and K Xue for comments and discussions, D Wilburn for assistance with ChIP experiments, J Andrie, J Akey, and D Gordon for Y12 and DBVPG6044 reference genomes and strains, and F Winston, G Sherlock, S Fields, C Payen, Y Zheng, and D Greig for strains.

## Additional information

### Funding

| Funder | Grant reference number | Author |
| --- | --- | --- |
| National Science Foundation | graduate research fellowship DGE-1256082 | Seungsoo Kim |
| National Institutes of Health | U54 DK107979 | William S Noble Jay Shendure |
| National Institutes of Health | P41GM103533 | William S Noble Maitreya J Dunham |
| National Institutes of Health | GM080484 | Jason H Brickner |
| Howard Hughes Medical Institute | | Jay Shendure |
| Howard Hughes Medical Institute | Faculty Scholar grant | Maitreya J Dunham |
| Canadian Institute for Advanced Research | Senior Fellow, Genetic Networks Program | Maitreya J Dunham |
| National Science Foundation | 1516330 | Maitreya J Dunham |

The funders had no role in study design, data collection and interpretation, or the decision to submit the work for publication.

### Author contributions

SK, Designed the study, Generated strains, Analyzed Hi-C experiments, Performed and analyzed ChIP and RNA-seq experiments, Wrote the paper, Assisted with preparing the manuscript; IL, Designed the study, Generated strains, Performed Hi-C experiments, Assisted with preparing the manuscript; DGB, Generated strains, Performed microscopy experiments; KC, Assisted with analysis of Hi-C data, Assisted with preparing the manuscript; WSN, Assisted with preparing the manuscript; JHB, Analyzed microscopy data, Assisted with preparing the manuscript; JS, MJD, Designed the study, Assisted with preparing the manuscript

### Author ORCIDs

Seungsoo Kim, http://orcid.org/0000-0002-5559-5289
Jason H Brickner, http://orcid.org/0000-0001-8019-3743
Jay Shendure, http://orcid.org/0000-0002-1516-1865
Maitreya J Dunham, http://orcid.org/0000-0001-9944-2666

## Additional files

### Supplementary files

• Supplementary file 1. Strains used in this study.

• Supplementary file 2. Primers used in this study.

• Supplementary file 3. Hi-C libraries.

### Major datasets

The following dataset was generated:

| Author(s) | Year | Dataset title | Dataset URL | Database, license, and accessibility information |
|---|---|---|---|---|
| Kim S, Liachko I, Brickner DG, Cook K, Noble WS, Brickner JH, Shendure J, Dunham MJ | 2016 | Data from The dynamic three-dimensional organization of the diploid yeast genome | https://www.ncbi.nlm.nih.gov/geo/query/acc.cgi?acc=GSE88952 | Publicly available at the NCBI Gene Expression Omnibus (accession no: GSE88952) |

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
