## [Decision Letter]

Thank you for submitting your article "The dynamic three-dimensional organization of the diploid yeast genome" for consideration by *eLife*. Your article has been reviewed by three peer reviewers, one of whom, Bing Ren (Reviewer #1), is a member of our Board of Reviewing Editors, and the evaluation has been overseen by Kevin Struhl as the Senior Editor.

The reviewers have discussed the reviews with one another and the Reviewing Editor has drafted this decision to help you prepare a revised submission.

1) There is general enthusiasm in this work, based on the innovative approach, the novel findings of sequence-dependent pairing of homologous chromosomes, and the rigorous experimental design.

2) There are several concerns that need to be addressed by further discussion and experiments as outlined below:

a) The evidence for HAS1 pairing due to interactions of both alleles with the nuclear pore complex is still indirect. It is recommended that the authors expand the in vivo imaging analysis to show how the pairing occurs in the 3D space of the nucleus, i.e. if the loci are only paired at the nuclear periphery.

b) To further strengthen the conclusion that nuclear pore complex mediates the pairing of HAS1 or TDA1 upon transcriptional activation, either of the following two experiments should be performed – 1) ChIP with a non-mobile nucleoporin (e.g. Nup60), or 2) localization of TDA1/HAS1 in a mutant that clusters pores (a nup133 truncation that lacks its N-terminus).

c) There is a lack of discussion of significance and relevance of the sequence-dependent homolog pairing observed here. It is recommended that the authors provide in-depth discussion on how pairing occurs in vivo, how this processes is facilitated by the NPC, whether the observed high frequency of pairing between HAS1 implies non-random, sequence dependent colocalization of the locus and NPC, and how transcription influence this process.

d) The authors should address the question of whether there are transcriptional changes in HAS1/TDA1 that accompany pairing and relocation to the nuclear periphery. A reviewer points out that SPELL groups TDA1 with carbohydrate metabolism genes, including HXK1 that goes to the nuclear pores in galactose, and whose expression changes upon glucose depletion.

[Editors' note: further revisions were requested prior to acceptance, as described below.]

Thank you for resubmitting your work entitled "The dynamic three-dimensional organization of the diploid yeast genome" for further consideration at *eLife*. Your revised article has been favorably evaluated by Kevin Struhl (Senior editor), a Reviewing editor, and two reviewers.

The manuscript has been improved but there are some remaining issues that need to be addressed before acceptance, as outlined below.

Reviewer #3:

The manuscript from Dunham and co-authors is significantly improved. They have shown that homologous chromosomes in diploid interact along their lengths more frequently than would be expected due to the Rabl-like configuration, particularly in saturated cultures. Their work also highlights the *HAS1-TDA1* alleles that dramatically associate in galactose and upon saturated growth. The association of these loci with the nuclear membrane and its dependence on Nup2 suggests that *HAS1-TDA1* pairing requires nuclear pore contact.

Major point

1) My only reservation is that the authors ought to include a sentence that acknowledges that association of *HAS1-TDA1* with nuclear pores has not yet been pinned down. In fact, the bulk of the evidence is negative or ambiguous: 1) Nup60 does not contact the genes, 2) nuclear pore contact does not require Mlp2, and 3) the only requirement for perinuclear enrichment is Nup2, which is a mobile nucleoporin that equilibrates on and off nuclear pores. Capelson et al., Cell, 2010 showed that mobile nucleoporins can play important roles at genes even when they are not at nuclear pores. It is acceptable for the authors to suggest that *HAS1-TDA1* interact with nuclear pores by a unique mechanism but they should also acknowledge that proof of nuclear pore contact is not yet definitive.

In this light, the Abstract should avoid the statement that the interaction is "dependent on interactions with nuclear pore complexes" and the title of the Figure 5 legend should not read, "requires nuclear pore interactions"

---

## [Author Response]

*The reviewers have discussed the reviews with one another and the Reviewing Editor has drafted this decision to help you prepare a revised submission.*

*1) There is general enthusiasm in this work, based on the innovative approach, the novel findings of sequence-dependent pairing of homologous chromosomes, and the rigorous experimental design.*

*2) There are several concerns that need to be addressed by further discussion and experiments as outlined below:*

*a) The evidence for HAS1 pairing due to interactions of both alleles with the nuclear pore complex is still indirect. It is recommended that the authors expand the* in vivo *imaging analysis to show how the pairing occurs in the 3D space of the nucleus, i.e. if the loci are only paired at the nuclear periphery.*

We have performed the suggested analysis by classifying our diploid cells, originally imaged for measuring distances between *HAS1-TDA1* homologs, based on their *HAS1-TDA1* peripheral localization (ON-ON, ON-OFF, or OFF-OFF, for the number of alleles at the periphery) and then counting the proportion which have paired *HAS1-TDA1* alleles (distance < 0.55 µm). Our results are shown in the new panel Figure 5. We find that *HAS1-TDA1* pairing is not restricted to the periphery.

However, pairing due to interactions with the nuclear pore need not result in pairing only at the nuclear periphery. Our previous work on *INO1* and *GAL1* show that at least some genes that interact with the nuclear pore upon activation move back to the nucleoplasm during S-phase (Brickner and Brickner, 2010) but remain paired in the nucleoplasm (Brickner et al., 2012). We reanalyzed our microscopy data from haploids to show that this cell cycle-dependent localization is true of *HAS1-TDA1* as well (Figure 5). Therefore, we believe that although the presence of *HAS1-TDA1* pairing in the nucleoplasm may suggest mechanisms for pairing in addition to nuclear pore interactions, this result does not rule out the role of nuclear pores in mediating pairing.

To clarify the role of nuclear pores in pairing, we have performed alternative analyses and experiments in addition to the cell cycle analysis described above, to provide additional evidence for the role of Nup2 in *HAS1-TDA1* pairing.

We had previously performed all peripheral localization analyses in haploids, for ease of deletion analyses. Thus, we also reanalyzed our diploid images to demonstrate that inducible peripheral localization also occurs in diploids (Figure 5).

We also acknowledge that in our initial submission, we had not described any direct evidence that Nup2 is required for pairing. Thus, to directly test whether Nup2 is required for both peripheral localization and pairing, we performed Hi-C in a *nup2Δ S. cerevisiae* x *S. uvarum* hybrid strain. Nup2 is in fact required for *HAS1-TDA1* pairing in galactose, but not in saturated culture. These results are illustrated in Figure 5. Given that *HAS1-TDA1* relocalizes to the nuclear periphery in saturated culture, *HAS1-TDA1* likely still interacts with nuclear pores, but may do so via alternative nuclear pore components. However, in light of these complicating results, we have revised our conclusions to weaken our claim that nuclear pores mediate *HAS1-TDA1* pairing.

*b) To further strengthen the conclusion that nuclear pore complex mediates the pairing of HAS1 or TDA1 upon transcriptional activation, either of the following two experiments should be performed – 1) ChIP with a non-mobile nucleoporin (e.g. Nup60), or 2) localization of TDA1/HAS1 in a mutant that clusters pores (a nup133 truncation that lacks its N-terminus).*

As suggested, we have performed ChIP for Nup60 in glucose and galactose, followed by both sequencing and qPCR. Although we see a clear enrichment for the positive control *GAL1* in galactose (Figure 5), indicating that the experiment worked, we do not see any enrichment in the *HAS1-TDA1* region (Figure 5). Although it is possible that the potentially indirect interaction of Nup60 with DNA is precluding detection of *HAS1-TDA1* enrichment, our observation of enrichment at *GAL1* indicates that at least, *HAS1-TDA1* does not interact with NPCs in the way that *GAL1* does. We believe this is consistent with the Mlp2-independence of *HAS1-TDA1* peripheral localization, unlike that of *GAL1* and *INO1*, and that it suggests *HAS1-TDA1* interacts with nuclear pores by a novel mechanism.

*c) There is a lack of discussion of significance and relevance of the sequence-dependent homolog pairing observed here. It is recommended that the authors provide in-depth discussion on how pairing occurs* in vivo*, how this processes is facilitated by the NPC, whether the observed high frequency of pairing between HAS1 implies non-random, sequence dependent colocalization of the locus and NPC, and how transcription influence this process.*

We have significantly expanded the Discussion section to describe the implications of our findings of homolog pairing both at the global and local scales. We discuss how the NPC may help mediate pairing by limiting the search space for the two homologs to find each other, and how interactions between DNA-binding proteins may lead to frequent colocalization of the homologous loci. We have also added a discussion of the relationship between pairing/relocalization and transcription.

*d) The authors should address the question of whether there are transcriptional changes in HAS1/TDA1 that accompany pairing and relocation to the nuclear periphery. A reviewer points out that SPELL groups TDA1 with carbohydrate metabolism genes, including HXK1 that goes to the nuclear pores in galactose, and whose expression changes upon glucose depletion.*

We thank the reviewer for the suggestion of investigating transcription of *TDA1*. We have performed RNA-seq to quantify transcriptional changes in galactose and saturated growth. As the reviewer suggested, we find that *TDA1* is upregulated in both galactose and saturated growth. This transcriptional pattern is consistent with the known biochemical function of Tda1 in phosphorylating Hxk2, the main hexokinase in yeast, and thereby leading to its export from the nucleus where it can interact with Mig1 to enact glucose repression. We have added Figure 6 to illustrate these results, and we discuss them at the end of the Results and also in the Discussion.

[Editors' note: further revisions were requested prior to acceptance, as described below.]

*Reviewer #3:*

*[…] Major point*

*1) My only reservation is that the authors ought to include a sentence that acknowledges that association of HAS1-TDA1 with nuclear pores has not yet been pinned down. In fact, the bulk of the evidence is negative or ambiguous: 1) Nup60 does not contact the genes, 2) nuclear pore contact does not require Mlp2, and 3) the only requirement for perinuclear enrichment is Nup2, which is a mobile nucleoporin that equilibrates on and off nuclear pores. Capelson et al., Cell, 2010 showed that mobile nucleoporins can play important roles at genes even when they are not at nuclear pores. It is acceptable for the authors to suggest that HAS1-TDA1 interact with nuclear pores by a unique mechanism but they should also acknowledge that proof of nuclear pore contact is not yet definitive.*

*In this light, the Abstract should avoid the statement that the interaction is "dependent on interactions with nuclear pore complexes" and the title of the Figure 5 legend should not read, "requires nuclear pore interactions"*

We have added a sentence to acknowledge the incomplete evidence for nuclear pore association and the need for further experiments to work out the mechanism. We have also revised the Abstract and the title of Figure 5 to clarify that our data suggest involvement of the nuclear pore complex but do not show its requirement.